# Inflammation Control and Immunotherapeutic Strategies in Comprehensive Cancer Treatment

**DOI:** 10.3390/metabo13010123

**Published:** 2023-01-13

**Authors:** Victor Ivanovich Seledtsov, Adas Darinskas, Alexei Von Delwig, Galina Victorovna Seledtsova

**Affiliations:** 1Innovita Research Company, 06116 Vilnius, Lithuania; 2Russian Scientific Center of Surgery Named after Academician B.V. Petrovsky, 119991 Moscow, Russia; 3National Cancer Institute, 08406 Vilnius, Lithuania; 4Institute for Fundamental and Clinical Immunology, 630099 Novosibirsk, Russia

**Keywords:** inflammation, anti-tumor immunity, pro-tumor immunity, immunotherapy, chemotherapy, cancer prognosis

## Abstract

**Highlights:**

**Abstract:**

Tumor growth and expansion are determined by the immunological tumor microenvironment (TME). Typically, early tumorigenic stages are characterized by the immune system not responding or weakly responding to the tumor. However, subsequent tumorigenic stages witness the tumor promoting its growth and metastasis by stimulating tumor-protective (pro-tumor) inflammation to suppress anti-tumor immune responses. Here, we propose the pivotal role of inflammation control in a successful anti-cancer immunotherapy strategy, implying that available and novel immunotherapeutic modalities such as inflammation modulation, antibody (Ab)-based immunostimulation, drug-mediated immunomodulation, cancer vaccination as well as adoptive cell immunotherapy and donor leucocyte transfusion could be applied in cancer patients in a synergistic manner to amplify each other’s clinical effects and achieve robust anti-tumor immune reactivity. In addition, the anti-tumor effects of immunotherapy could be enhanced by thermal and/or oxygen therapy. Herein, combined immune-based therapy could prove to be beneficial for patients with advanced cancers, as aiming to provide long-term tumor cell/mass dormancy by restraining compensatory proliferation of surviving cancer cells observed after traditional anti-cancer interventions such as surgery, radiotherapy, and metronomic (low-dose) chemotherapy. We propose the *Inflammatory Prognostic Score* based on the blood levels of C-reactive protein and lactate dehydrogenase as well as the neutrophil-to-lymphocyte ratio to effectively monitor the effectiveness of comprehensive anti-cancer treatment.

## 1. Introduction

*Intra vitam*, the immune system combats infectious agents and controls reparative cell growth, thus effectively sustaining integrity and cohesiveness of the body. Of relevance, anti-infectious and anti-tumor immunities are fundamentally based on the same innate and adaptive mechanisms. However, as opposed to infections, tumors are portrayed as “wounds that do not heal” [1] in that they hijack the proliferative resolution phase in the wound healing and repair process to encourage tumor progression. In other words, the immune system tends to recognize tumors as a self-tissue that requires regeneration, with subsequent provision of a supportive tumor microenvironment (TME) for tumor growth and expansion. Hence, basic innate and adaptive immune mechanisms underlying tumor progression are generally similar to those accounting for the regenerative activity in normal tissues, which are tightly controlled in acute wounds and are grossly imbalanced in tumors [2].

Tumor fate is believed to be defined by a fine balance between anti-tumor versus pro-tumor immunity. Innate anti-tumor immunity relies on classically activated (N1) neutrophils and macrophages (M1), natural killer (NK) cells, natural killer T (NKT) cells, and γ/δ T cells. NK cells recognize stress ligands on tumor cells with reduced or absent MHC expression. NKT cells use NK surface receptors and the invariant CD1d-restricted T-cell receptor (TCR) to recognize glycolipids, while γδ T cells recognize tumor-derived phosphoantigens or stress ligands by engaging semi-invariant γ/δ TCR [3]. Adaptive anti-tumor immunity is principally based on mature dendritic cells (DCs), macrophages, tumor-specific CD4+ T helper 1 cells (Th1), CD8+ cytotoxic T cells (CTLs), and cytotoxic antibody (Ab)-producing B cells. In this paradigm, professional antigen (Ag), presenting cells DCs and macrophages, cleave endogenous and exogenous Ags to small antigenic peptides for binding to, and presentation by, major histocompatibility (MHC) class I and class II molecules, resulting in the clonal expansion of CD8+ and CD4+T cells, respectively. Principal T cells with anti-tumor activity (i.e., Th1 cells) are characterized by the ability to promote cell-mediated CD8+ cytotoxic T lymphocyte (CTL) generation, classical macrophage (M1) and neutrophil (N1) activation as well as the activation of other effector cells with cytotoxic potential such as NK cells, NKT cells, and γδ T cells [3,4].

Pro-tumor innate immunity is based on alternatively activated neutrophils (N2) and (M2) macrophages, myeloid-derived suppressor cells (MDSCs), and innate regulatory T (Tregs) and B cells (Bregs), whereas adaptive immunity is mediated via adaptive Tregs and Th2 cells. Th2 cells facilitate the activation of M2 macrophages, N2 neutrophils, and myeloid-derived suppressor cells (MDSCs) [3,4], and have the ability to directly and indirectly suppress the functional activity of Th1 cells and vice versa, which allows for the sustained polarization of immune reactivity [5].

IL-17-secreting Th17 cells have been reported to possess both pro-tumor and anti-tumor activity [6], thus constituting a conundrum with respect to their role in cancer immunity. As far as B cells are concerned, tumor-specific B cells differentiating into cytotoxic antibody (Ab)-secreting plasma cells comprise an important constituent of anti-tumor immune reactivity. In contrast, regulatory B cells produce immunosuppressive cytokines such as transforming growth factor beta (TGF-β) and IL-10, thus effectively downregulating immune (including anti-cancer) reactivity [3].

Table 1 demonstrates bipolarity of tumor growth regulation by cellular and humoral factors, which is further complicated by the fact that some cell types (such as neutrophils and macrophages) could reveal their yin and yang faces (i.e., serving both as anti-tumor and pro-tumor guards depending on particular conditions). Some soluble factors follow suit in this respect, as exemplified by reactive oxygen species (ROS), which on one hand, could mediate cytotoxic activity of neutrophils and macrophages, and on the other hand, suppress T cell reactivity [7,8]. Phospholipases constitute another example of functional dichotomy being able to directly exterminate tumor cells, while also participating in the destruction of inter-tissue barriers, thereby promoting tumor invasion [3]. Even pro-inflammatory interferon-γ (IFN-γ) activity could furnish bidirectional effects with respect to tumor growth by inducing and maintaining cellular anti-tumor immunity, while also downregulating anti-tumor immunity via IFN-γ-mediated stimulation of checkpoint molecule expression [9,10] and upregulation of indoleamine 2,3-dioxygenase-1 (IDO-1) activity [11,12]. Bifunctionality of IL-2, a well-known T cell growth and differentiation factor, reveals itself in stimulating Th1 cells and CTL with anti-tumor activity, but at the same time, serving as a potent activator of Tregs, which inhibits immune-mediated anti-tumor activity [3]. On one hand, granulocyte-macrophage colony-stimulating factor (GM-CSF) is a pivotal molecule in the generation of mature DCs that trigger adaptive immune responses, but on the other hand, it is also an inductor and activator of immunosuppressive MDSCs [3,4].

Anti-tumor immunity is far less diversified and robust, so that even one defective component may lead to a dramatic deterioration of the entire anti-tumor protection. In contrast, tumor protective immune mechanisms are constituents of normal robust regenerative machinery that accounts for bodily integrity with abundant feedback and back-up units, so that a single faulty component is unlikely to result in a major system failure.

## 2. The Role of Inflammation in Tumor Development

Tumors can be viewed from the immunotherapeutic and clinical perspective as immunologically “cold”, “warm”, “hot”, and “excessively hot” based on the in situ immune cell infiltration and production of soluble mediators regulating tumor growth (Figure 1). Lack or paucity of immune cell infiltration in the tumor bed and low immunogenicity of tumor-associated Ags are the main characteristics of “cold” tumors. Although “warm” (or immune-excluded) tumors are also devoid of immune cells attracted to the tumor bed, there is some degree of immune cell infiltration in the surrounding stroma. A full range of cancer evasion strategies from immune surveillance employed by “cold” and “warm” tumors at early disease stages allows for an initially stealthy development of the tumorigenic process. However, upon reaching certain quantitative tumor burden characteristics, “cold” and “warm” tumors do engage in interactions with the immune system. In particular, “hot” (or immune-inflamed) tumors are characterized by higher tumor immunogenicity and better anti-tumor immune responses, principally due to: (i) significant accumulation of immune cells with cytotoxic and cytostatic activity, with immune cells appearing adjacent to tumor cells [9,13]; (ii) the expression of immune cell-attracting molecules such as CCL5 and CXCL9 [14]; and (iii) a type I interferon (IFN) transcriptional signature with important implications for cancer cell growth control [15].

Inducing local inflammatory conditions in tumors (i.e., converting “cold” and “warm” tumors into “hot” ones is a pivotal task of immunotherapeutic treatments. One should bear in mind, however, that a tumor tends to sustain local inflammation in order to reprogram the immune system toward supporting regenerative processes and converting to an “excessively hot” tumor. We maintain that such “overheating” tumors are counterproductive in terms of anti-tumor immunity. Indeed, “overheating” tumors are common at advanced disease stages, being marked by significant inflammatory infiltrates due to the substantial damage incurred to the tumor and surrounding normal tissues. In these settings, cell destruction releases large amounts of damage-associated molecular patterns (DAMPs) into circulation, which, as intrinsic danger signals, are sensitized and recognized by pattern recognition receptors (PRR) to stimulate pro-tumor (regenerative) immunity [16]. “Overheating” tumors due to disproportionate tumor-driven inflammation can thwart the effectiveness of immunotherapeutic interventions. Indeed, inflammation, on one hand, attracts anti-tumor immune cells to TME, and on the other hand, excessive inflammation creates favorable conditions for tumor growth and metastasis, notably by upregulating the expression of checkpoint inhibitory molecule expression and promoting the migration and functionality of pro-tumor immune cells [4]. Therefore, combination therapy of advanced cancers should necessarily include anti-inflammatory therapy conducive to “cooling down” exaggerated tumor-mediated inflammation and to dampen pro-tumor immune activity [17] by drugs such as cyclooxygenase-2 (COX-2) inhibitors, H2-blockers, phosphodiesterase-5 (PDE-5) inhibitors, statins, activator of AMP-activated protein kinase, and glucocorticoids [4,16]. Importantly, all of these drugs are widely used in the clinic and could be applied for combined anti-cancer therapy.

Below, we provide a brief overview of various anti-cancer immunotherapeutic platforms designed to boost anti-tumor defense (or “heat”, but not “overheat” tumors) to achieve the holy grail of immunotherapy (i.e., programming the immune system of the patient to control cancer). We also summarize alternative strategies devised to suppress pro-tumor immunity.

## 3. Boosting Anti-Tumor Immunity

Various therapeutic monoclonal antibody (mAb)-based modalities constitute an essential treatment strategy for a variety of disorders including cancer. Current bioengineering technologies allow for the development of low-immunogenic humanized cytotoxic IgG mAbs of desired specificity, with several anti-cancer mAb-based technological platforms existing on the biopharmaceutical market. Moreover, anti-tumor immune responses are commonly boosted by mAb-based inactivation of immunosuppressive checkpoint molecules [PD-1/PD-L1, cytotoxic T lymphocyte-associated antigen-4 (CTLA-4), lymphocyte activation gene-3 (LAG-3), T cell immunoglobulin and mucin-domain containing-3 (TIM-3), T cell immunoglobulin and ITIM domain (TIGIT), V-domain Ig suppressor of T cell activation (VISTA), etc.], which are known to suppress the activation and function of T cells and downregulate immune (including anti-cancer) reactivity [4,18]. However, while PD-1/PD-L1 and CTLA-4 immune checkpoint molecule blockade holds great promise as an effective immunotherapeutic approach to treat cancer in experimental and clinical conditions displaying different degrees of success in some cancers including melanoma, non-small cell lung carcinoma (NSCLC), and urothelial carcinoma, many reports have shown low or no responsiveness in other cancer types such as gastrointestinal, breast, pancreatic, prostate, sarcoma, and colorectal cancers [9]. In these cases, anti-cancer immune responses could be revitalized by T-cell activation agonists, with tumor necrosis factor receptor superfamily (TNFRSF) members regarded as principal targets for Ab-based immunostimulatory cancer immunotherapy [4].

Anti-tumor immune reactivity has also been shown to be subject to cytokine (type I and II IFNs, IL-2, IL-7, IL-12, IL-15, IL-18, and IL-21) regulation and augmentation in preclinical and clinical settings. In particular, enhancement of the tumoricidal activity of different lymphocyte effector cells (such as NK cells) and anti-tumor polarization of other immune cells (such as N1 neutrophils, M1 macrophages, and Th1 cells) could be achieved by the application of pro-inflammatory cytokines [3,4,5,15,19].

Anti-cancer immunity and tumor rejection phenomena could be boosted by using activators of Toll-like receptor (TLR) as well as cytoplasmic NOD-like receptors (NLRs) and retinoic acid inducible gene 1-like receptors (RLRs), which are currently considered as important targets for immunotherapy against cancer such as myeloid malignancies. For example, ligating intracellular TLR (3, 7, 8, or 9) could force DCs and macrophages to increase the production of IFN-α, IL-6, IL-8, IL-12, and TNF-α, thus enhancing Th1 cell-mediated anti-cancer immune responses [20,21].

Age is an important factor in selecting particular immunotherapy regiments, as age is associated with a decline in the immune system (immunosenescence), and thus inevitably with an increased cancer susceptibility. Of importance, immune-mediated anti-tumor reactivity in elderly patients can be boosted, for instance, by melatonin-dependent enhancement of Th1-mediated responses [22] or thymic peptides that facilitate T-cell activation and survival [4].

Different types of cancer vaccines including whole tumor cell-based and protein/peptide vaccines as well as RNA and DNA vaccines have been developed in the last few years [3,4]. According to our clinical experience, xenogeneic cell-based vaccines could be also an effective tool to activate anti-tumor immune responses in cancer patients. Indeed, xenogeneic proteins have been shown to be much more efficacious in breaking the immune tolerance to tumor compared to homologous analogs [23]. Xenogeneic vaccine technology entails the application of readily available animal tumor cell lines, thus ensuring maximal antigenic overlap with target tumors to achieve desired anti-tumor immune responses.

Another promising anti-tumor Ag-nonspecific immunotherapeutic approach consists of adoptive cell immunotherapy (ACI), which involves isolation, activation, and expansion of particular autologous immune cells with their subsequent reintroduction into the patient’s bloodstream or TME. This technological platform operates independently of MHC-dependent restriction, being primarily formulated on the basis of NK and NKT cells, cytokine-induced killer cells (CIKs), macrophage activated killer cells, etc., (i.e., cells with pronounced broad-spectrum tumoricidal activity) [24,25]. Ag-specificity within the ACI platform could be achieved by introducing tumor-infiltrating lymphocyte (TIL) or genetically modified T cells such as TCR engineered T cells (TCR-T) and chimeric Ag receptor-engineered T cells (CAR-T) that recognize tumor-associated Ags in a MHC-dependent and independent manner [26]. We emphasize that long-term cultivation of immune cells could lead to the selection of cell clones characterized by a high sensitivity to apoptosis and low responsiveness to immunoregulation signals. Administration of such immune cells with profoundly defective immune functions could be counterproductive and result in the development of serious immune-mediated diseases [26].

Our recent R&D efforts have focused on developing a leukocyte-based technology based on the transfusion of leukocytes treated with immunostimulatory agents for a relatively short period of time (≤20 h). This treatment has been found to dramatically increase functional activity of leukocytes, including T lymphocytes [27]. Elderly people with weak and compromised immune systems constitute the major cohort within the entire population affected by oncological diseases. Therefore, we reasoned that this immunotherapeutic platform could be supplemented with the administration of cells from healthy young donors. Indeed, cancer patients with chemotherapy-induced neutropenia and infections have previously been shown to benefit from allogeneic leukocyte transfusions. Such immunotherapeutic approaches based on leukocytic preparations derived from young healthy donors have been shown to possess high anti-tumor activity against advanced-stage solid tumors, with minimal side effects observed [28]. In these settings, allogeneic NK cells could have higher anti-cancer activity compared to autologous NK cells [29]. Interestingly, ACI procedures based on allogeneic donor cells were effectively amalgamated with low-dose nonmyeloablative chemotherapy to boost graft-versus-tumor (but not graft-versus-host) reactivity [30,31]. In this scenario, graft-versus-tumor effect was mediated by donor-derived T cells, which were also shown to engage recipient immune cells into long-term anti-tumor reactivity [32]. According to our experience, ACI protocols utilized in elderly people could benefit from the administration of allogeneic cells from a closely related person. We hypothesized that such allogeneic cells would serve as potent stimuli of autologous NK cell-mediated cytotoxic activity, while similar HLA haplotypes would be instrumental in developing adaptive anti-tumor reactivity.

## 4. Suppressing Pro-Tumor Immunity

As we alluded to above, tumor destiny is controlled by a fine equilibrium between anti-tumor versus pro-tumor immunity. From this perspective, it is clear that suppressing pro-tumor immune reactivity would strengthen anti-tumor responses. This can be achieved by inhibiting immunosuppressive soluble factors such as the IDO-1 enzyme characterized by high levels of expression in the TME of various tumors [3,4]. Similarly, depleting suppressive cells such as Tregs (with anti-CD25 mAbs) have also been shown to enhance anti-tumor immunity, leading to tumor rejection [33]. In the same vein, resiquimod (an inhibitor of Treg function and TLR7/8 agonist) showed clinical effects in treating some cancers [34]. MDSCs are considered the “queen bee” of TME, being a key factor that shields tumors from host immune responses. This situation can be reversed, for example, by (i) all-trans-retinoic acid (ATRA), which directly forces MDSCs to differentiate into mature macrophages and DCs; (ii) anti-inflammatory triterpenoids, which downregulate MDSC-dependent immunosuppressive activity [35]; (iii) class I histone deacetylase inhibitors (entinostat), which neutralize MDSCs through epigenetic reprogramming [36]; or (iv) various approved pharmacological agents (e.g., PDE5 inhibitors, COX-2 inhibitors, arginase 1 inhibitors, bisphosphonates, gemcitabine, and paclitaxel) that directly or indirectly inhibit MDSC-mediated suppressive activity [3].

## 5. Role of Immunotherapy in Multimodal Multipurpose Treatment of Cancer

Simultaneous targeting of different multidirectional effectors and regulatory mechanisms in the context of combination immunotherapy is more likely to surmount cancer resistance by aiming (i) to control tumor-driven inflammation; (ii) enhance anti-tumor immunity; and (iii) downregulate pro-tumor immunity. Indeed, compared with the individual treatment alone, higher effectiveness and overall survival were demonstrated upon combining different anti-cancer immunotherapies in patients with breast and lung cancer [4,37]. In addition, the combined application of different immunotherapeutics allows for the reduction in individual dose regimens, thus minimizing the side effects [38].

There is a clear possibility of combining immunotherapy with traditional first-line treatments of cancer patients such as surgical tumor removal at early disease stages. However, we stress that surgery also damages normal tissues and induces the release of DAMPs into the bloodstream, which could induce tumor protective inflammation and provoke the development of residual disease. This situation indicates immunotherapy application to prevent tumor relapses in a post-operative period to be potentially combined with thermal and/or oxygen therapy, which are known to enhance the activity of immune anti-cancer mechanisms. Indeed, moderate hyperthermia has been shown to drastically augment the anti-cancer activity of immune effector cells [39], while oxygen therapy is capable of selectively destroying tumor cells by inducing oxidative stress in tumor cells as well as to downregulate suppressive TME-mediated effects by preventing TME acidification [40].

Chemotherapy remains an important intervention strategy in combined anti-cancer therapy. However, we maintain that chemotherapy application at early disease stages could result in the early prevalence of drug-resistant tumor populations, thus underlying subsequent chemotherapy failure at advanced disease stages when cytoreduction is of utmost necessity. With this in mind, we envisage that immunotherapy should become a mainstream therapeutic platform in combined anti-tumor treatment protocols at early disease stages (Table 2).

At advanced cancer stages, chemoradiotherapy causes damage not only to the tumor, but also to normal tissues, thus levying additional strain on all adaptive regenerative mechanisms including that of immune origin. We stress that the tumor is equipped with pathological regenerative capacity, so that the partial destruction of tumor cells generates many more tumor cells (Figure 2). Adaptive capacities of tumor cells are known to increase during the oncological process, resulting in diminished sensitivity to cytoreductive treatments, thus preventing the elimination of all tumor cells at advanced disease stages [41]. Based on our experience, we propose that the anti-tumor immunotherapy of patients at advanced disease stages should aim to reconstitute and stimulate immune mechanisms that control tumor cell growth and the execution of regenerative tumor activity. Figure 2 shows that following cytoreduction not accompanied by immunotherapy, the tumor is subjected to pro-tumor immune cell activity. Therefore, the principal aim of immunotherapy consists in attracting anti-tumor immune cells in TME to take tumor growth under effective immunological control. In these settings, immunotherapy could receive support from low-dose chemotherapy (metronomic) to optimize the effectiveness of tumor growth control, rather than achieve tumor cell destruction (Table 2). Metronomic chemotherapy-dependent augmentation of tumor immunogenicity has been shown to be dependent on ensuing cellular death, improved Ag cross-presentation in lymph nodes, and an increased generation of Ag-experienced T cells [9]. Of particular interest in this regard is the clinical application of hypomethylating agents such as decitabine and guadecitabine, which enhance the expression of cancer/testis Ags in tumor cells and significantly increases tumor susceptibility to anti-tumor Ag-specific immune mechanisms [42]. Moreover, chemotherapy could curb pro-tumor inflammation as well as promote the formation of niches in lymphoid tissues, facilitating the expansion of anti-tumor immune cells [4].

We maintain that achieving cancer dormancy, rather than complete tumor destruction, should be the principal aim of cancer patient treatment at advanced disease stages. We defined cellular dormancy as transitory G_0_–G_1_ growth arrest occurring in some cancer cells. The proposed attenuated “balanced coexistence” strategy for treating patients at advanced disease stages could be more rational and beneficial to prolong the patients’ lives.

Despite the high incidence of tumor dormancy, the underlying molecular programs are not clearly understood. According to our data, tumor cells resistant to contact-dependent interactions remain sensitive to contact-dependent inhibition by normal cells including that of immune origin [43,44,45]. Therefore, the initiation of tumor growth is supported primarily by reciprocal tumor cell-dependent (but not tumor-normal cell-dependent) contacts. We envisage that it is a pronounced immune cell infiltration of the tumor that is the major factor limiting tumor growth, wherein immune cells control tumor growth via contact-dependent intercommunications and the production of cytostatic/cytotoxic factors [46]. In agreement, tumor infiltration by immune cells, and most notably T cells, is considered to be a positive prognostic factor [13].

Subject to anti-tumor immune defense deterioration, and in particular due to increased inflammation in TME, tumor dormancy has been shown to be reversible, thus providing an opportunity for the tumors to enter the progression state again. Therefore, while tumor-driven inflammation promotes metastatic outgrowth, the inclusion of anti-inflammatory agents into combined anti-cancer regimens could interfere with dormant cell reawakening, thus drastically reducing the risk of metastatic relapses [12].

## 6. Prognostic Effectivity Assessment of Combined Anti-Tumor Therapy Protocols

The efficacy of combined anti-tumor immunotherapeutic treatments requires objective monitoring to ensure timely treatment protocol amendments, paying particular attention to clinical pseudoprogression and pseudoregression phenomena. Indeed, tumor loci can become more conspicuous during various instrumental imaging studies due to inflammation, which could indicate disease progression in some cases. Tumor burden can increase after combined therapy, leading to some patients experiencing delayed tumor shrinkage. Based on these examples, pseudoprogression defines primary tumor increase or new lesion appearance after tumor regression, which histologically is accompanied by immune cell infiltration and recruitment of immune cells to the tumor [47]. Alternatively, inflammation reduction caused by chemotherapy could make tumor lesions less visible, which could be interpreted as pseudoregression. Therefore, the interpretation of instrumental imaging data from patients in receipt of immunotherapeutic treatments should be considered with caution [47].

We maintain that the inflammatory reactivity of the immune system determines the fate of the tumor, so that markers of inflammation provide effective insight into true cancer progression or regression. On the one hand, local anti-tumor inflammatory responses could constitute a positive prognostic factor, while on the other hand, non-specific systemic tumor-driven inflammation associated with cancer progression and normal tissue damage indicates a negative prognosis. Based on the pivotal role of inflammation in cancer prognosis assessment, we selected three of the most important inflammatory markers, which, importantly, have been previously described to have a prognostic value in their own right. To make the assessment of cancer prognosis more accurate, we developed a novel “Inflammatory Prognostic Score” (IPS) based on the following blood parameters: C-reactive protein (CRP), lactate dehydrogenase (LDH), and neutrophil-to-lymphocyte ratio (NLR) [4]. To the practical clinician, IPS provides a summary assessment of inflammation, tissue damage, and immune cellular reactivity. The proposed scoring system is in agreement with multiple clinical studies that demonstrated a strong association of upregulated inflammation markers with poor disease prognosis. In the proposed IPS scale, each marker is assessed according to the binary numeral system consisting of 0 (normal) or 1 (increased). The prognosis is determined by summing up all three values, which is consistent with four possible outcome prediction scores: 0—good; 1—doubtful; 2—moderately bad; and 3—bad prognosis (Table 3).

The proposed IPS scale contributes to the further development of the Glasgow Prognostic Score (GPS) based on blood albumin and CRP levels, which is traditionally used in the clinic to predict survival outcomes in various cancer types [48]. We obtained solid clinical evidence that the proposed IPS scoring system correlates with the Glasgow prognostic score results. Moreover, IPS-based assessment allows for more efficient clinical analysis to introduce/amend anti-inflammatory and immunotherapeutic protocols. We hypothesize that our proposed IPS scale will find its place in controlling inflammatory reactivity observed during tumorigenic process and/or after the administration of certain immunotherapeutics including immune checkpoint inhibitors and CAR-T. By all means, the IPS scoring system suggested herein needs further extensive assessment in clinical practice to prove its usefulness.

## 7. Conclusions

Immunotherapeutic interventions are progressively gaining a footing in the vast field of anti-tumor treatment at all stages of the disease. At early stages, anti-tumor immunotherapy facilitates the elimination of tumor cells and small post-operative metastases, thus effectively preventing the development of residual disease. At advanced stages, combined immunotherapy could be applied to stimulate the immunological mechanisms involved in tumor growth control. We propose inflammation modulation as a pivotal factor in cancer immunotherapy. Importantly, immunotherapy can be effectively combined with all traditional cytoreductive therapeutic interventions, thus permitting the reduction in anti-cancer dosage regiments as well as minimizing the side effects. The proposed Inflammatory Prognostic Scale based on three parameters (CRP, LDH, and NLR) could prove to be useful for the timely introduction of modifications in the multidirectional comprehensive treatment of cancer.

## Figures and Tables

**Figure 1 metabolites-13-00123-f001:**
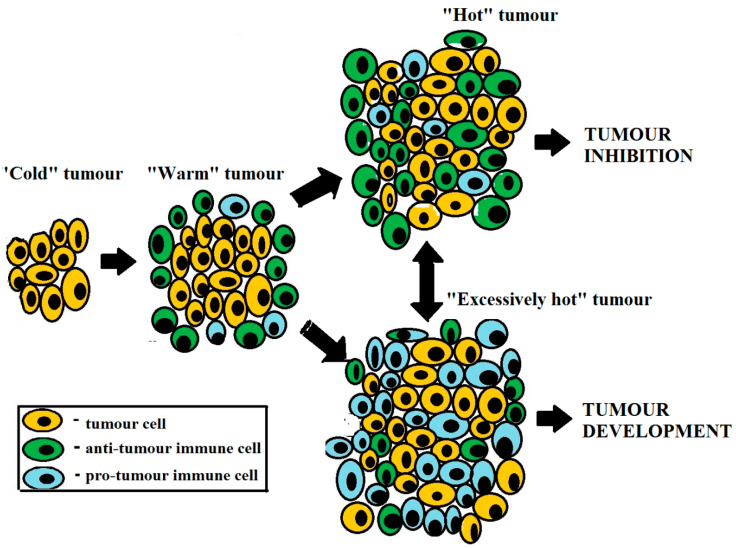
Types of cancer from the immunological and immunotherapeutic perspective. Tumors are considered as immunologically “cold”, “warm”, “hot”, or “excessively hot” based on immune cell infiltration.

**Figure 2 metabolites-13-00123-f002:**
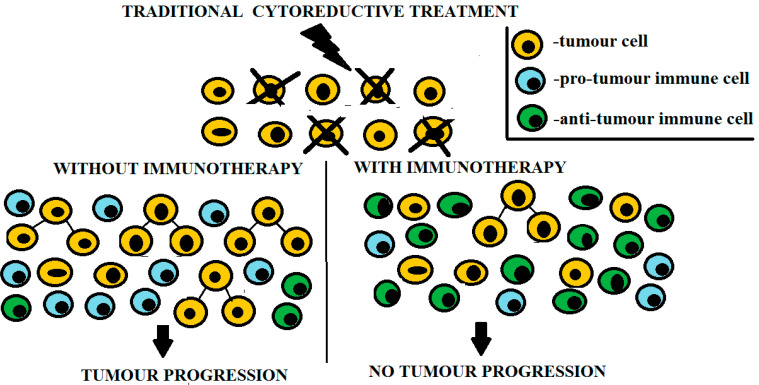
Tumor growth-controlling effects of immunotherapy-driven inflammation in patients with advanced cancer. Immunotherapy-induced infiltration of tumor by anti-tumor immune cells provides effective control over tumor growth. However, if cytoreductive treatment is not accompanied by immunotherapy, the tumor is subjected to predominant pro-tumor immune cell activity.

**Table 1 metabolites-13-00123-t001:** Cellular and humoral factors involved in anti-tumor vs. pro-tumor downstream immune-mediated effects.

Anti-Tumor Activity	Pro-Tumor Activity
**Cells**	
N1 neutrophil; M1 macrophage; natural killer (NK) cell; NKT (natural killer T) cell; γ/δ T cell; classical mature dendritic cell (DC); Th (T helper) 1 cell; Th17 cell; cytotoxic T lymphocyte (CTL); B cell.	N2 neutrophil; M2 macrophage; myeloid-derived suppressor cell (MDSC); tolerogenic immature DC; Th2 cell; Th17 cell; regulatory T cell (Treg); regulatory B cell (Breg).
**Soluble factors**	
Tumor necrosis factor (TNF); IFN (interferon)-γ; type I IFNs; IL (interleukin)-2; IL-7; IL-15; IL-18; granulocyte-macrophage colony-stimulating factor (GM-CSF); reactive oxygen species (ROS); proteases, extracellular adenosine triphosphate (eATP); phospholipases.	Transforming growth factor (TGF-β); IL-2; IL4; IL6; IL10; GM-CSF; granulocyte colony-stimulating factor (G-CSF); vascular endothelial growth factor VEGF); indoleamine 2,3-dioxygenase-1 (IDO-1); proteases; prostaglandin E (PGE); ROSs; phospholipases; histamine; adenosine (ADO).
Effects	
Inhibition of neovascularization, downregulation of pro-tumor immunity, tumor inhibition.	Stimulation of neovascularization, downregulation of anti-tumor immunity, tumor promotion.

Notes. The data presented is not intended to cover all known and putative cellular and humoral factors with tumor growth modulatory activity, and only proven factors relevant to this review are shown.

**Table 2 metabolites-13-00123-t002:** Possible combined anti-cancer treatments at the early and advanced disease stages.

Disease Stage	Possible Treatments	Results
Local tumor (I–II stage)	Surgery; immunotherapy; hyperthermia; oxygen therapy.	Recovery
Advanced tumor (III–IV stage)	Surgery; immunotherapy; hyperthermia; oxygen therapy; radiotherapy; chemotherapy.	Tumor growth inhibition; prolongation of patient’s life

**Table 3 metabolites-13-00123-t003:** Inflammatory Prognostic Score.

Parameter	Normal	Increased
C-reactive protein (CRP)	0	1
Lactate dehydrogenase (LDH)	0	1
Neutrophil-to-lymphocyte ratio (NLR)	0	1

Notes: Inflammatory Prognostic Score interpretation: 0—good; 1—doubtful; 2—moderately bad; 3—bad prognosis.

## Data Availability

The data presented in this study are available in the main article.

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
