# Peer review of "Inflammation Control and Immunotherapeutic Strategies in Comprehensive Cancer Treatment"

_metabolites, 2023, doi:10.3390/metabo13010123_

Round 1

Reviewer 1 Report

The paper is interesting, however it needs some ipmrovement before it may be accepted for publicaiton:

Line 341 - names of the drugs should not be written with capital letters, 

Figure 2 - "See the description in 356 the text." whould be replaced with brief summary of the photos - all of them should be self-explanatory.

Lines 379, 393, 402, 414 - text size is different and should be unified. 

All the photos and tables should have the abbreviations explained in their descirptions.

The IPS scale seems misterious from the description of the Authors. Why they chose only 3 parameters and why exactly these parametrs would give us any insight into the tumor. Both CRP and LDH are systemic markers and may be elevated due to other than cancer conditions. Please add a presice description of the proposed scale and rationale behind it. 

Also please add some general idea of the paper to the beginning. For me the whole structure of the paper is confusing - first part looks like a review paper and then the Authors move to the IPS scale which seems to the most important part of the paper, but is bearly described in the manuscript. In my opinion, the Authors shoudl focus on the scale and make it the most prominent part of the paper. 

Author Response

Reviewer 1

The paper is interesting, however it needs some ipmrovement before it may be accepted for publicaiton:

  1. Line 341 - names of the drugs should not be written with capital letters,
  2. Figure 2 - "See the description in the text." whould be replaced with brief summary of the photos - all of them should be self-explanatory.
  3. Lines 379, 393, 402, 414 - text size is different and should be unified.
  4. All the photos and tables should have the abbreviations explained in their descirptions.
  5. The IPS scale seems misterious from the description of the Authors. Why they chose only 3 parameters and why exactly these parametrs would give us any insight into the tumor. Both CRP and LDH are systemic markers and may be elevated due to other than cancer conditions. Please add a presice description of the proposed scale and rationale behind it.
  6. Also please add some general idea of the paper to the beginning. For me the whole structure of the paper is confusing - first part looks like a review paper and then the Authors move to the IPS scale which seems to the most important part of the paper, but is bearly described in the manuscript. In my opinion, the Authors shoudl focus on the scale and make it the most prominent part of the paper.

Our responses

We are grateful to the reviewer for the invaluable comments and suggestions, which allowed us to critically review our MS and amend it accordingly.

  1. In the revised MS the names of drugs are written in lowercase letters.
  2. As suggested by the reviewer, we expanded and amended Figure 2 notes section to read: “Tumour growth-controlling effects of immunotherapy in patients with advanced cancer. Immunotherapy-induced infiltration of tumour by anti-tumour immune cells provides effective control over tumour growth. However, if cytoreductive treatment is not accompanied by immunotherapy tumour is subjected to predominant pro-tumour immune cell activity.”. The statement "See the description in the text" has been removed, as suggested.
  3. The same font size of the MS has been adjusted throughout the text.
  4. All abbreviations present in Figures and Tables have been explained, as per the reviewer’s suggestion.
  5. The key concept of our view point paper states that the inflammatory reactivity of the immune system determines the fate of the tumour. Therefore, blood inflammatory markers are of pivotal importance in terms of tumour prognosis. Our extensive clinical experience in cancer immunotherapy allowed us to select three most important inflammatory markers, which, importantly, have been previously described to have a prognostic value in its own right. We went further with the idea of developing a novel most accurate cancer prognostic tool based on individual inflammatory markers by combining those three inflammatory markers in the context of a single prognostic scale. We reasoned that the combined prognostic power of these markers within a single Inflammatory Prognostic Scale (IPS) would make the assessment of cancer prognosis much more accurate. Currently, we use IPS scale in the test mode to accumulate clinical data in patients with advanced prostate cancer, and preliminary data is very encouraging. As the IPS scale is of a universal value and could be applied in various cancer settings, we decided to incorporate information about our approach to cancer prognosis in this paper to prompt professional oncologists to test it in the clinic.

As suggested by the reviewer, we added key statements marked in red in the revised MS in order to explain the rationale behind the proposed IPS scale. To strengthen our argumentation basis, we also added the following statement to the Conclusions section: “We propose inflammation modulation as a pivotal factor in cancer immunotherapy”.

  1. We stress that our MS is not a traditional review, but rather a novel viewpoint that emphasises the pivotal role of inflammation control in a successful anti-cancer immunotherapy strategy. The description of this novel concept requires a certain intrinsic logic that broadly speaking follows the following argument sequence (and, hence, sections of the MS): tumour › inflammation › inflammation modulation by immunotherapy › clinical effect › clinical prognosis and IPS scale. We made significant efforts to describe the rationale behind the proposed IPS scale in the relevant section of the revised MS.

As suggested by the reviewer, we added the key statement marked in red in the Abstract section of the revised MS.

Reviewer 2 Report

The manuscript by Seledtsov et al. presents a rather superficial description of the role of the immune system in the evolution of tumors as well as the role of immunotherapy. In general, the content of the manuscript does not add anything new to the literature published so far.

In particular, in reference to inflammation, mentioned both in the title and in the abstract as one of the important points of the manuscript, it only appears in the final part and without any connection with the previous content. There is no methodology of any kind and no conclusions are drawn. 

Author Response

Reviewer 2

The manuscript by Seledtsov et al. presents a rather superficial description of the role of the immune system in the evolution of tumors as well as the role of immunotherapy. In general, the content of the manuscript does not add anything new to the literature published so far.

In particular, in reference to inflammation, mentioned both in the title and in the abstract as one of the important points of the manuscript, it only appears in the final part and without any connection with the previous content. There is no methodology of any kind and no conclusions are drawn.

Our responses

We are grateful to the reviewer for the opinion on our work. We reiterate that our MS represents a view point paper, and its structure and logic are coherent with the idea of inflammation modulation as a pivotal factor in cancer immunotherapy. Generally speaking, we think we used the appropriate structure, argumentation and body of references to support our view point. Indeed, our extensive clinical experience provided solid support for this notion, which prompted us to summarise this novel approach in the context of a scientific publication. We appreciate that various professionals may have different experience giving rise to different medical concepts, which will be equally interesting for the highly regarded readership of Metabolites.

We also stress that there are numerous excellent traditionally structured reviews in the open literature (including those originating from our laboratory, ref 3, 4 ) that contain detailed analysis of anti-cancer immune mechanisms. Our approach was completely different, in that we focussed on summarising our extensive clinical and laboratory experience in the context of a View Point paper. This format allowed us to formulate the following novel ideas in the area of cancer immunotherapy, such that we do not agree with the reviewer that “the manuscript does not add anything new to the literature published so far”.

  1. In particular, our hypothesis required us to focus on the balance between anti-tumour versus pro-tumour immune mechanisms, which allowed us to define for the first time a particular advanced tumour state as an “excessively hot”, i.e. when tumour promotes its own growth by actively facilitating favourable conditions via stimulating pro-tumour (regenerative) immunity.
  2. Another novel feature of this paper constitutes in describing a firm linkage between inflammation and cancer disease prognosis.
  3. Furthermore, we put forward an original idea that rational anti-cancer therapy at advanced disease stages should aim at long-term tumour growth suppression, rather than tumour extermination.

Reviewer 3 Report

In this review the Authors summarize the main anti-tumor immunological mechanisms as well as the role of inflammation in cancer development. The Authors also explain the rationale of potential  immunotherapeutic strategies (inflammation modulation, antibody-based immunostimulation, drug-mediated immunomodulation, cancer vaccination, adoptive cell immunotherapy) that might be synergistically applied in cancer patients to amplify and achieve more effective anti-tumour immune responses. Moreover, the Authors elaborate and propose an Inflammatory Prognostic Score, based on blood levels of C-reactive protein and lactate dehydrogenase, as well as neutrophil-to-lymphocyte ratio, in monitoring the effectiveness of comprehensive anti-cancer treatment.

I think this is a good work, which might be useful in clinical practice; moreover, the linear scholarly exposition of the manuscript will be also useful for students approach.

Author Response

Reviewer 3

In this review the Authors summarize the main anti-tumor immunological mechanisms as well as the role of inflammation in cancer development. The Authors also explain the rationale of potential immunotherapeutic strategies (inflammation modulation, antibody-based immunostimulation, drug-mediated immunomodulation, cancer vaccination, adoptive cell immunotherapy) that might be synergistically applied in cancer patients to amplify and achieve more effective anti-tumour immune responses. Moreover, the Authors elaborate and propose an Inflammatory Prognostic Score, based on blood levels of C-reactive protein and lactate dehydrogenase, as well as neutrophil-to-lymphocyte ratio, in monitoring the effectiveness of comprehensive anti-cancer treatment.

I think this is a good work, which might be useful in clinical practice; moreover, the linear scholarly exposition of the manuscript will be also useful for students approach.

Our responses

We would like to express our gratitude to the reviewer for kind words about our work. We agree that publications of this kind would prove to be interesting and useful for clinicians, academic professionals and students alike, i.e. the highly renowned readership of Metabolites.

Reviewer 4 Report

This manuscript by Seledtsov et al. described a pioneering point of view regarding the fundamental role of inflammation control in effective cancer treatment. In this manuscript, the authors provided a comprehensive overview of the immune cells involved in cancer immunity and highlighted the pro-tumor effect of inflammation, leading to the proposal of the Inflammatory Prognostic Score (IPS) in the design of cancer treatment regimens. Overall, this manuscript presents an inspiring viewpoint of translating inflammation control for treating cancer. Comments on this manuscript are listed as follows.

Major:

1.      The authors claim that the aim of cancer treatment in the future is to half tumors to the dormancy state instead of tumor eradication. However, the advantages and disadvantages of this argument still need to be fully elaborated. The authors are encouraged to make a stronger argument by enlisting more supporting lines of evidence.

2.      Likewise, the idea about the translational potential of the Inflammatory Prognostic Score (IPS) can be more substantiated by more lines of supporting evidence.

3.      How to combine inflammation control with current immunotherapy options, such as immune checkpoint inhibitors and CAR-T, to achieve more effective cancer treatment? Are there any supporting reports to exemplify this proposal?

Minor:

1.      Line 76, the comma in the sentence “Th2 cells, are characterized by: (i)…” should be deleted.

2.      Table 1 is suggested to start at the beginning of page 3 for easier reading.

3.      Line 351, the number presented in “G0-G1” should be in the subscript form.

Author Response

Reviewer 4

This manuscript by Seledtsov et al. described a pioneering point of view regarding the fundamental role of inflammation control in effective cancer treatment. In this manuscript, the authors provided a comprehensive overview of the immune cells involved in cancer immunity and highlighted the pro-tumor effect of inflammation, leading to the proposal of the Inflammatory Prognostic Score (IPS) in the design of cancer treatment regimens. Overall, this manuscript presents an inspiring viewpoint of translating inflammation control for treating cancer. Comments on this manuscript are listed as follows.

Major:

  1. The authors claim that the aim of cancer treatment in the future is to half tumors to the dormancy state instead of tumor eradication. However, the advantages and disadvantages of this argument still need to be fully elaborated. The authors are encouraged to make a stronger argument by enlisting more supporting lines of evidence.
  2. Likewise, the idea about the translational potential of the Inflammatory Prognostic Score (IPS) can be more substantiated by more lines of supporting evidence.
  3. How to combine inflammation control with current immunotherapy options, such as immune checkpoint inhibitors and CAR-T, to achieve more effective cancer treatment? Are there any supporting reports to exemplify this proposal?

Minor:

  1. Line 76, the comma in the sentence “Th2 cells, are characterized by: (i)…” should be deleted.
  2. Table 1 is suggested to start at the beginning of page 3 for easier reading.
  3. Line 351, the number presented in “G0-G1” should be in the subscript form.

Our responses

We are thankful to the reviewer for detailed analysis of our work.

  1. Achieving cancer dormancy rather than aiming to tumour eradication is one of the important points that we discuss in the MS based on available literature and our own experience. However, we appreciate that this idea is in a rather embryonic state, and therefore this paper is marked as a Viewpoint.
  2. Background and rationale behind the proposed IPS scale was elaborated in more detail while working with comments expressed by Reviewer No. 1.
  3. Clinical application of the proposed IPS scale for inflammation control is one of the major points of this MS. To emphasise this notion, we added the following statement: “We hypothesise that our proposed IPS scale will find its place in controlling inflammatory reactivity observed during tumorigenic process and/or after administration of certain immunotherapeutics, including such as immune checkpoint inhibitors and CAR-T.”

All minor points were also corrected in accordance to the reviewer’s suggestions.

Round 2

Reviewer 1 Report

The authors have revised the manuscript and now I hope it may be accepted for publication.

Author Response

Thank you veru much.

Reviewer 2 Report

I agree with the authors' response. 

Author Response

Thank you veru much.